# Xianling Gubao attenuates high glucose-induced bone metabolism disorder in MG63 osteoblast-like cells

Xinlong Chen[1,2◉], Yan Li[3◉], Zhongwen Zhang[1], Liping Chen[4], Yaqian Liu[4], Shuhong Huang[5], Xiaoqian Zhang[1]*

1 Shandong Key Laboratory of Rheumatic Disease and Translational Medicine, Department of Endocrinology and Metabolism, The First Affiliated Hospital of Shandong First Medical University & Shandong Provincial Qianfoshan Hospital, Shandong Institute of Nephrology, Jinan, China, 2 Hospital of Chengdu University of Traditional Chinese Medicine, Chengdu, Sichuan, China, 3 Shandong Medicine and Health Key Laboratory of Clinical Pharmacy, Department of Clinical Pharmacy, The First Affiliated Hospital of Shandong First Medical University & Shandong Provincial Qianfoshan Hospital, Shandong Engineering and Technology Research Center for Pediatric Drug Development, Jinan, China, 4 Department of Endocrinology and Metabolism, Weifang Medical University, Shandong Provincial Qianfoshan Hospital, Weifang, China, 5 Institute of Basic Medicine, The First Affiliated Hospital of Shandong First Medical University, Jinan, Shandong Province, China

◉ These authors contributed equally to this work.
* zhxqqy@163.com

**Data Availability Statement:** All materials and data in this study are available at the following website (https://figshare.com/articles/dataset/Xianling_

## Abstract

Diabetes mellitus (DM) patients are prone to osteoporosis, and high glucose (HG) can affect bone metabolism. In the present study, we investigated the protective effects of traditional Chinese herbal formulation Xianling Gubao (XLGB) on HG-treated MG63 osteoblast-like cells. MG63 cells were incubated with control (mannitol), HG (20 mM glucose) or HG + XLGB (20 mM glucose+200 mg/L XLGB) mediums. Cell proliferation, apoptosis, migration and invasion were examined using CCK8, colony-formation, flow cytometry, Hoechst/PI staining, wound-healing and transwell assays, respectively. ELISA, RT-PCR and western blot analysis were used to detect the levels of osteogenesis differentiation-associated markers such as ALP, OCN, OPN, RUNX2, OPG, and OPGL in MG63 cells. The levels of the PI3K/Akt signaling pathway related proteins, cell cycle-related proteins, and mitochondrial apoptosis-related proteins were detected using western blot analysis. In HG-treated MG63 cells, XLGB significantly attenuated the suppression on the proliferation, migration and invasion of MG63 cells caused by HG. HG downregulated the activation of the PI3K/Akt signaling pathway and the expressions of cell cycle-related proteins, while XLGB reversed the inhibition of HG on MG63 cells. Moreover, XLGB significantly reduced the promotion on the apoptosis of MG63 cells induced by HG, the expressions of mitochondrial apoptosis-related proteins were suppressed by XLGB treatment. In addition, the expressions of osteogenesis differentiation-associated proteins were also rescued by XLGB in HG-treated MG63 cells. Our data suggest that XLGB rescues the MG63 osteoblasts against the effect of HG. The potential therapeutic mechanism of XLGB partially attributes to inhibiting the osteoblast apoptosis and promoting the bone formation of osteoblasts.

Gubao_zip/20393118) or DOI:10.6084/m9.
figshare.20393118.

**Funding:** This work was supported by Shandong
traditional Chinese Medicine Science and
Technology Development Plan Project (No.2017-
169), Shandong Province medical and health
science and technology development plan project
(NO. 2017WS617), the Key Research &
Development Plan of Shandong Province (No.
2018GSF118176), Natural Science Foundation of
China (82000788) and Natural Science Foundation
of Shandong Province (No. ZR2016HQ26).

**Competing interests:** The authors declare that
there are no conflicts of interest.

## Introduction

Diabetes mellitus (DM) is becoming one of the most important diseases worldwide with an estimated 451 million people with diabetes worldwide in 2017, and is expected to increase to 693 million by 2045 [1]. Sustained high blood glucose levels can lead to extensive vascular damage and altered bone metabolism, resulting in various complications, such as increased risk of osteoporosis [2–4]. The main effects of DM on bone metabolism are the increase of bone resorption and the decrease of bone formation, which result in the decrease of bone mineral content and osteoporosis and the increase of fracture risk [5]. However, the mechanism of diabetic osteoporosis is more complicated. Compared with non-diabetic patients, the bone mineral density of patients with diabetes is reduced [6,7]. The osteoporosis treatment of diabetic patients is the same as that of non-diabetic patients, including drug treatment, physical therapy intervention, and surgical treatment [8]. The application of drugs in the treatment of osteoporosis is still a clinical challenge, since some conventional drugs, such as Calcium, vitamin D, bisphosphonates and calcitonin, promote the risk of cardiovascular events [9–11]. Hence, it is necessary to find a new treatment for diabetic osteoporosis. Emerging evidence has shown that traditional Chinese herbal medicine and Chinese herbal formulas have a broad prospect in the treatment of diabetic osteoporosis [12,13].

Xianling Gubao (XLGB) is a traditional Chinese herbal formulation approved by the China Food and Drug Administration (CFDA; China, Z20025337) and has been used for the treatment of osteoporosis, osteoarthritis, aseptic osteonecrosis, and postmenopausal osteoporosis for more than 20 years in China [14,15]. XLGB is composed of six herbs with percentages in weight: Epimedium brevicornu Maxim (70%), Dipsacus inermis Wall (10%), Cullen corylifolium (L.) Medik. (5%), Anemarrhena asphodeloides Bunge (5%), Salvia miltiorrhiza Bunge (5%), and Rehmannia glutinosa (Gaertn.) DC (5%) [16]. It has been reported that XLGB has a preventive effect on ovariectomized (OVX)-induced bone loss in mice; its bioactive compounds promote the proliferation and/or mineralization of osteoblast-like UMR 106 cells [17,18]. However, whether XLGB is effective for osteoporosis induced by high glucose is undetermined.

The aim of the present study was to assess the effect of XLGB on diabetic osteoporosis. We investigated the effect of XLGB on the cellular function of high glucose induced-human osteoblast-like cells MG63 in vitro and explored its mechanism.

## Materials and methods

### Chemicals

The stock solution (mother liquor) was prepared from crude drug as following: the powder in XLGB capsules (0.5g × 4; Guizhou Tongjitang Pharmaceutical Co., Ltd. batch number 171204) were dissolved in 10mL deionized water and incubated for 2h at room temperature. After centrifugation (3000 g for 20 min), the supernatant was filtered through a 0.22 μm filter to a final concentration of 200 mg/ml (mother liquor). Since flavonoids have been shown to be the main active components in Epimedium brevicornu Maxim, we determined the major flavonoid compounds epimedin C and Icariin content by high-performance liquid chromatography (HPLC). The contents of Epimedin C and Icariin in the stock solution were 0.816 mg/mL and 0.196 mg/mL, respectively. The HPLC chromatograms were shown in **Fig 1**. The reference substance of Epimedin C (111780–201905, purity: 94.3%) and Icariin (110737–201516, purity: 94.2%) were purchased from China Food and Drug Testing Institute., The mother liquor of XLGB leaching solution (200 mg/ml) was diluted 1000 times for subsequent experiment before use, so the working concentration of XLGB was 200mg/L.

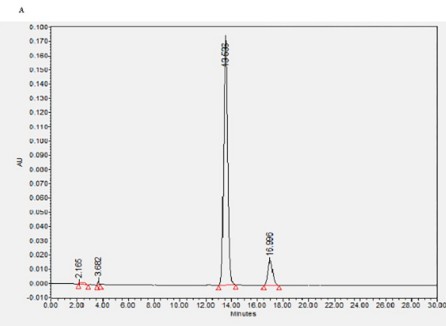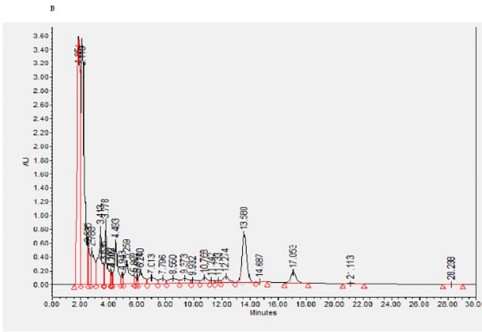

**Fig 1. The contents of Epimedin C and Icariin in XLGB powder were determined by HPLC.** A. HPLC chromatograms of the reference substance of Epimedin C and Icariin B. HPLC chromatograms of XLGB powder. 1. Epimedin C, 2. Icariin.

## Cell culture and treatment

Human osteoblast-like cells MG63 were cultured in DMEM (Dulbecco's Modified Eagle Medium) containing 10% fetal bovine serum (FBS; Thermo Fisher Scientific, USA) and 100 mg/mL penicillin-streptomycin (Invitrogen, USA) at 37°C with 5% $CO_2$. Cells were treated with high glucose (20 mM glucose; HG) or HG + XLGB (20 mM glucose + 200 mg/L XLGB), respectively; mannitol (20 mM) was used as a control.

## CCK8 assay

Cell Counting Kit-8 (CCK8) assay was performed to analyze cell viability. For the dose-dependent experiments, MG63 cells were seeded in 96-well plates at a density of 1000 cells/well. Cells were incubated for 48 h with different supplementation of glucose (0, 2.5, 4.0, 5.0, 7.5, 10.0, 15.0, 20.0, 30.0, 40.0, and 50.0 mM) + different concentrations of XLGB (0, 25, 50, 75, 100, 150, 200, 400, 600, 800, 1000, 1500, and 2000 mg/L) mediums. Following by incubation with CCK8 regent (10 μl per well) at 37°C for 1.5 h, the OD value of each well was measured at 450nm using a microplate reader. For cell viability assay, cells were seeded in 96-well plates and cultured with mannitol (control), HG (20 mM) or HG (20 mM) + XLGB (200 mg/L) mediums for 0, 24, 48 and 72 h, respectively. The OD value of each group was measured according to the above methods.

## Colony formation assay

MG63 cells were plated in 35 mm dishes (1000 cells/dish) and incubated with mannitol (control), HG (20 mM) or HG (20 mM) + XLGB (200 mg/L) mediums at 37°C for 2–3 weeks, respectively. When the visible colony appeared, the culture was terminated, and cells were washed twice with PBS. Cells were then fixed with 4% paraformaldehyde for 20 min and stained with crystal violet for 30 min. The number of colonies was counted.

## Flow cytometry assay

An Annexin V-FITC and PI apoptosis detection kit (Solarbio, Beijing, China) was used to measure cell apoptosis. Following by incubation with mannitol, HG (20 mM) or HG (20 mM) + XLGB (200 mg/L) mediums at 37°C for 24 h, MG63 cells were cultured with serum free medium for another 24 h. The cells ($5 \times 10^5$) were stained with 5 μl of Annexin V-FITC in the dark at room temperature for 5 min, and stained with PI for another 30 min. Samples were

analyzed by a flow cytometry (BD FACSC Anto II, BD Biosciences, USA), and calculated using BD FACSDiva software (BD Bioscience).

## Hoechst/PI staining

MG63 cells were plated in a 96-well plate at a density of 1000 cells/well and incubated with mannitol, HG (20 mM) or HG (20 mM) + XLGB (200 mg/L) mediums at 37˚C for 48 h. Then, cells were stained with Hoechst 33342/PI (10μg/mL) at 37˚C for 15 min. After immersing twice with PBS, the fluorescence of each group was observed under a fluorescence microscope (Leica, Germany).

## Western blot analysis

After being incubated with mannitol, HG (20 mM) or HG (20 mM) + XLGB (200 mg/L) mediums for 48 h, MG63 cells were lysed using RIPA buffer. Protein concentration of each group was measured using a BCA Protein Assay (Solarbio). 20 μg protein of each group was separated by SDS-PAGE and then transferred to a PVDF membrane (Millipore, MA, USA) followed by blocking with 5% nonfat milk for 1 h. After being incubated with the primary antibodies (Bcl-2, Bax, active Caspase 3, Akt, p-Akt, p70S6K, CDK4, CDK6, Cyclin D1 and GAPDH; Cell Signaling Technology, USA) overnight at 4˚C, the membranes were then incubated with secondary antibodies (Cell Signaling Technology) at room temperature for 1 h. The bound antibodies were visualized with an ECL kit (Solarbio) and analyzed by Quantity One software. GAPDH was used as an internal reference, and the relative expression of proteins was normalized to control.

## ELISA assay

The levels of alkaline phosphatase (ALP), Osteocalcin (OCN), Osteopontin (OPN), runt-related transcription factor 2 (RUNX2), Osteoprotegerin (OPG), and Osteoprotegerin ligand (OPGL) in cell supernatant samples of each group were examined by ELISA kit (Sigma-Aldrich, Germany) according to the manufacturer's instructions. Briefly, MG63 cells were cultured in mannitol, HG (20 mM) or HG (20 mM) + XLGB (200 mg/L) mediums for 48 h, and cell supernatant was then collected. After being blocked with 5% BSA overnight at 4˚C, the pre-coated microplates were incubated with serial dilutions of standard or cell supernatant of each group. Following by incubation of 2 h at room temperature, the microplates were then incubated with the horseradish peroxidase-conjugated polyclonal antibodies for 40 min at room temperature. The OD value was measured at 450 nm after the cells were incubated with substrate solution for 20 min.

## RT-PCR analysis

Total RNA was extracted from each group cells using Trizol (CWBIO, Beijing, China), and cDNA was synthesized using the HiFiScript cDNA Synthesis Kit (CWBIO). Real-time PCR reaction was performed to measure the mRNA levels of target genes using the SYBR Fast qPCR Mix (Takara). β-actin was used as internal reference. The obtained data were analyzed using the $2^{-\Delta\Delta Ct}$ method, and the relative expression levels of genes were normalized to control group.

## Wound-healing assay

Cells were plated in a 6-well plate at a density of $5\times10^5$ cells/well and cultured in medium overnight. A scratch was generated with a tip and cells were rinsed with PBS. Following by

incubation with mannitol, HG (20 mM) or HG (20 mM) + XLGB (200 mg/L) mediums without FBS for 0, 12, and 24 h, cells were captured. The percentage of wound closure was analyzed using ImageJ software.

### Transwell assay

Transwell chambers pre-coated with Matrigel were performed for cell invasion assay, no Matrix gel chambers were used for migration assay. Cells incubated with mannitol, HG (20 mM) or HG (20 mM) + XLGB (200 mg/L) mediums for 24 h were harvested and suspended in serum-free medium. Cells ($1 \times 10^4$) were added in the upper chambers, and the lower chambers were filled with DMEM medium supplemented with 10% FBS. Following by 24 h of incubation, the non-invading or non-migrating cells were removed by scrubbing. The invaded or migrated cells were fixed with 4% paraformaldehyde for 30 min and stained with 0.1% crystal violet for another 20 min. The number of stained cells was counted in five randomly selected microscopic fields under the microscope.

### Statistic analysis

All data were recorded as mean ± standard deviation and analyzed using GraphPad Prism 7. Student's t-test or ANOVA analysis was used for comparison between two or more groups. $P < 0.05$ was considered statistically significant.

## Results

### XLGB rescues the MG63 cells against the effect of high glucose on proliferation

First of all, we evaluated the effective concentration of the main components of XLGB. Considering XLGB is a kind of compound proprietary Chinese Medicine, and the concentrations of its main ingredients Icariin and Epimedin C are often used for researches instead of the mixture powder. Flavonoids have been shown to be the main active components in Epimedium brevicornu Maxim, therefore, we determined the major flavonoid compounds Epimedin C and Icariin content by HPLC. The epimedin C and Icariin concentration in the mother liquor of XLGB were 0.816 mg/ml and 0.196 mg/ml, respectively. We diluted the mother liquor of XLGB leaching solution (200 mg/ml) by 1000 times before use, so the working concentration of Epimedin C and Icariin was 0.816 mg/L and 0.196 mg/L respectively.

To assess the effect of high glucose on the proliferation of osteoblasts cells, MG63 cells were cultured with osteogenic induction medium containing different concentrations of glucose for 48 h. As indicated by CCK8 assay, 15 mM or higher supplementation of glucose significantly inhibited the viability of MG63 cells in a dose-dependent manner, and the $IC_{50}$ of glucose was 20 mM (**Fig 2A**). Similarly, MG63 cells were treated with different concentrations of XLGB in high glucose medium to investigate its effect on cell viability. As shown in **Fig 2B**, when the concentration of XLGB was less than or equal to 200 mg/L, there was no significant effect on the viability of MG63 cells, but when its concentration was 400 mg/L or higher, it showed significant inhibition. Therefore, 20 mM of glucose and 200 mg/L of XLGB were used in the following experiments due to their appropriate inhibition. Subsequently, we found that treatment with XLGB significantly reversed the inhibition of high glucose on the viability of MG63 cells compared with the HG group (**Fig 2C**). Colony formation assay was performed to further verify. Results showed that the number of colonies formed in the HG group was significantly decreased compared with the control group, but XLGB could abolish this inhibitory effect of high glucose (**Fig 2D**).

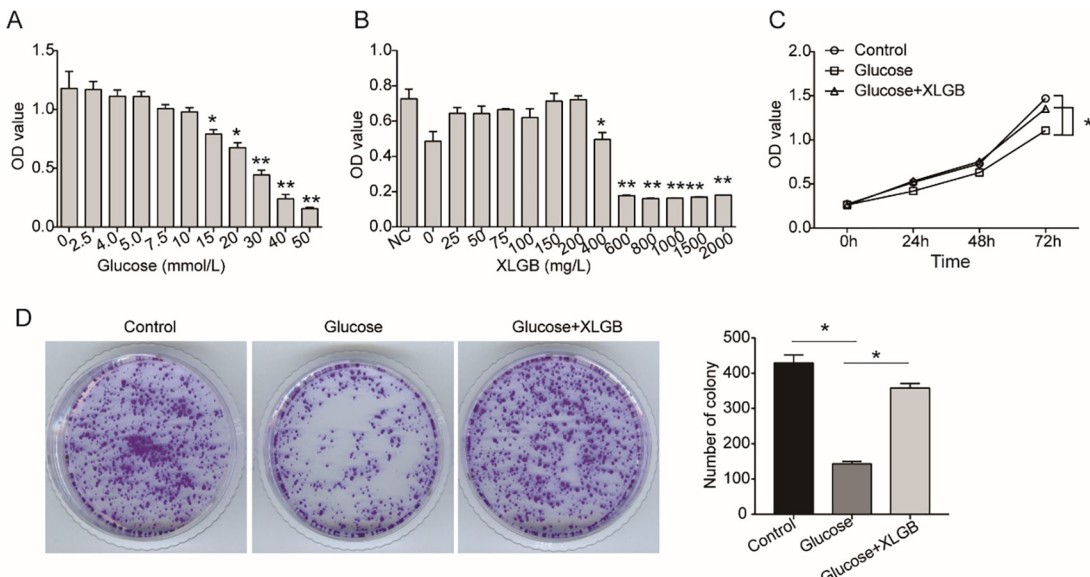

**Fig 2. XLGB rescues the inhibitory effect of high glucose on the proliferation of MG63 cells.** A. CCK8 assay was performed to assess the effect of different concentrations of glucose on the proliferation of MG63 cells. B. Effect of different concentrations of XLGB on the proliferation of MG63 cells exposed to high glucose (20 mM). C. Cell proliferation was examined by CCK8 assay as described in the methods at 0, 24, 48 and 72h. D. Colony formation assay was used to assess cell clonality. Control, cells treated with mannitol; Glucose, cells treated with 20 mM of glucose; Glucose +XLGB, cells treated with 20 mM of glucose plus 200 mg/L of XLGB. Data represented mean ± SD of three separate experiments. *$P<0.05$, **$P<0.01$.

## XLGB rescues the MG63 cells against the effect of high glucose on apoptosis

Flow cytometry analysis showed that high glucose obviously increased the percentage of apoptotic MG63 cells compared with the control group, while the addition of XLGB eliminated the effect of high glucose on apoptosis (**Fig 3A**). Additionally, Hoechst/PI staining assay also confirmed that high glucose promoted apoptosis of MG63 cells, while XLGB could resist the promoting effect of high glucose on apoptosis of MG63 cells (**Fig 3B**). Further, the expression levels of apoptosis-related proteins were examined to investigate the mechanism of XLGB on apoptosis of MG63 cells. As indicated by western blot analysis, we found that the expression of Bcl-2 down-regulated by high glucose was rescued by XLGB in MG63 cells, the expression levels of Bax and active Caspase 3 up-regulated by high glucose were also rescued by XLGB (**Fig 3C**). Collectively, these data suggest that high glucose promotes apoptosis of MG63 cells by regulating the Bcl-2/Bax axis and Caspase 3, while XLGB can enhance the resistance of MG63 cells to high glucose.

## XLGB rescues high glucose-reduced expression of osteoblast differentiation-associated markers in MG63 cells

We detected the expression levels of osteoblast differentiation-associated markers (ALP, OCN, OPN and RUNX2) [19] using ELISA assay to investigate the effect of high glucose and XLGB on the differentiation of MG63 cells. We observed that high glucose significantly reduced the level of ALP in MG63 cells, while XLGB abolished the downregulation of ALP by high glucose (**Fig 4A**). Similarly, another three differentiation-associated markers, OCN, OPN and RUNX2 were also downregulated in the HG-treated MG63 cells, which were rescued by XLGB (**Fig 4B–4D**). In addition, the expression of OPG was significantly inhibited by high glucose in

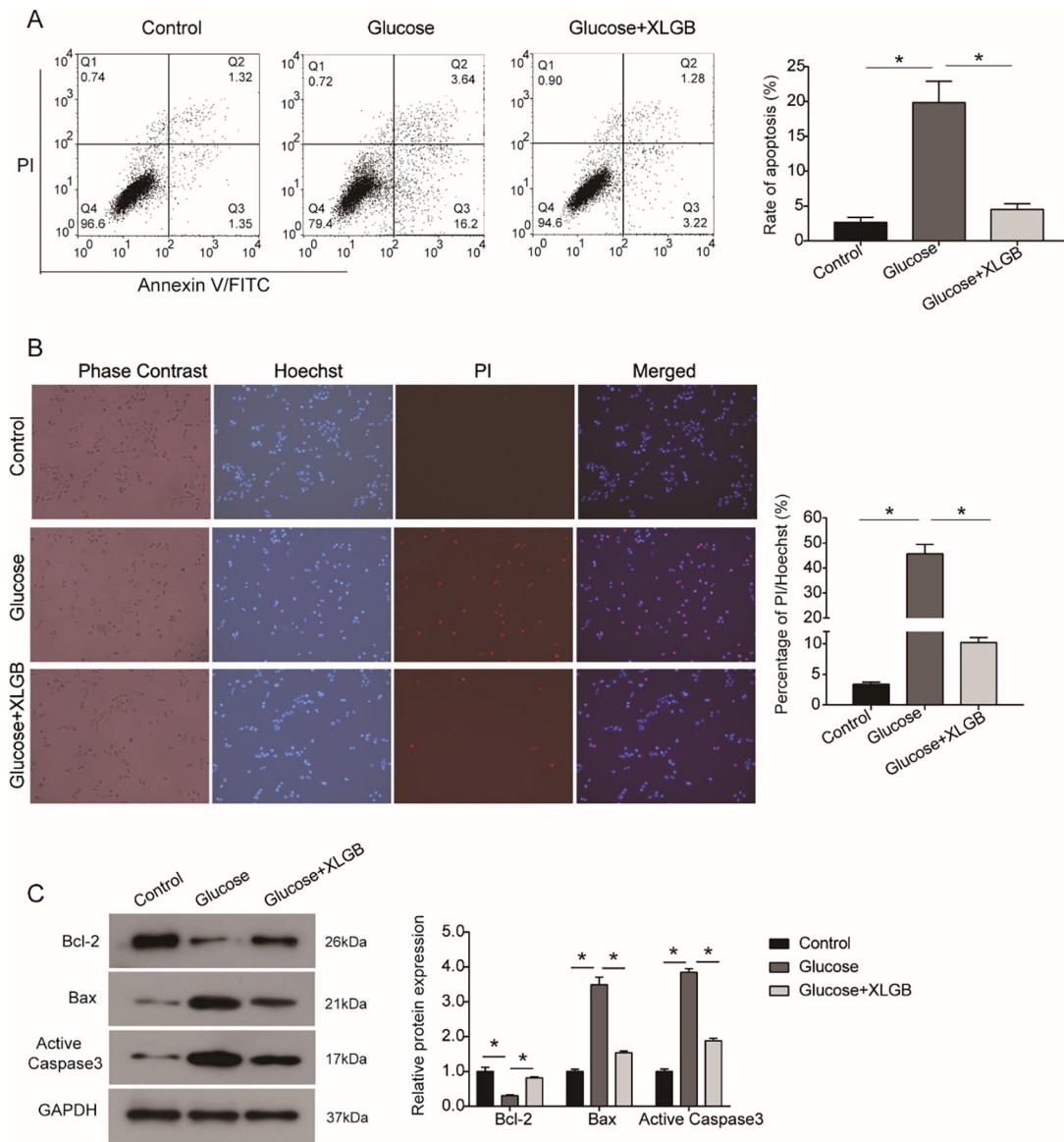

**Fig 3. XLGB rescues the promotion of high glucose on apoptosis of MG63 cells.** A. Flow cytometry was performed to examine the percentage of apoptotic MG63 cells incubated in control, high glucose or high glucose +XLGB mediums. B. Hoechst/PI staining of MG63 cells after indicated treatment. C. The relative expressions of apoptosis-related proteins Bcl-2, Bax and active Caspase 3 were detected by western blot. Control, cells treated with mannitol; Glucose, cells treated with 20 mM of glucose; Glucose +XLGB, cells treated with 20 mM of glucose plus 200 mg/L of XLGB. Data represented mean ± SD of three separate experiments. $^*P<0.05$.

MG63 cells, while the expression of OPGL was up-regulated; the regulation of high glucose on the expression of OPG and OPGL was relieved by XLGB (**Fig 4E and 4F**). To reconfirm such effects of XLGB on the differentiation of MG63 cells in high glucose, we then detected the mRNA levels of these markers. It was demonstrated that the down-regulated mRNA expressions of ALP, OCN, OPN, and RUNX2 by high glucose were rescued by XLGB in MG63 cells (**Fig 4G**). Additionally, consistent with the ELISA results, the decrease in OPG mRNA expression and the increase in OPGL mRNA expression induced by high glucose were both restored by XLGB (**Fig 4H**). The results of western blot also showed that inhibitions of differentiation-

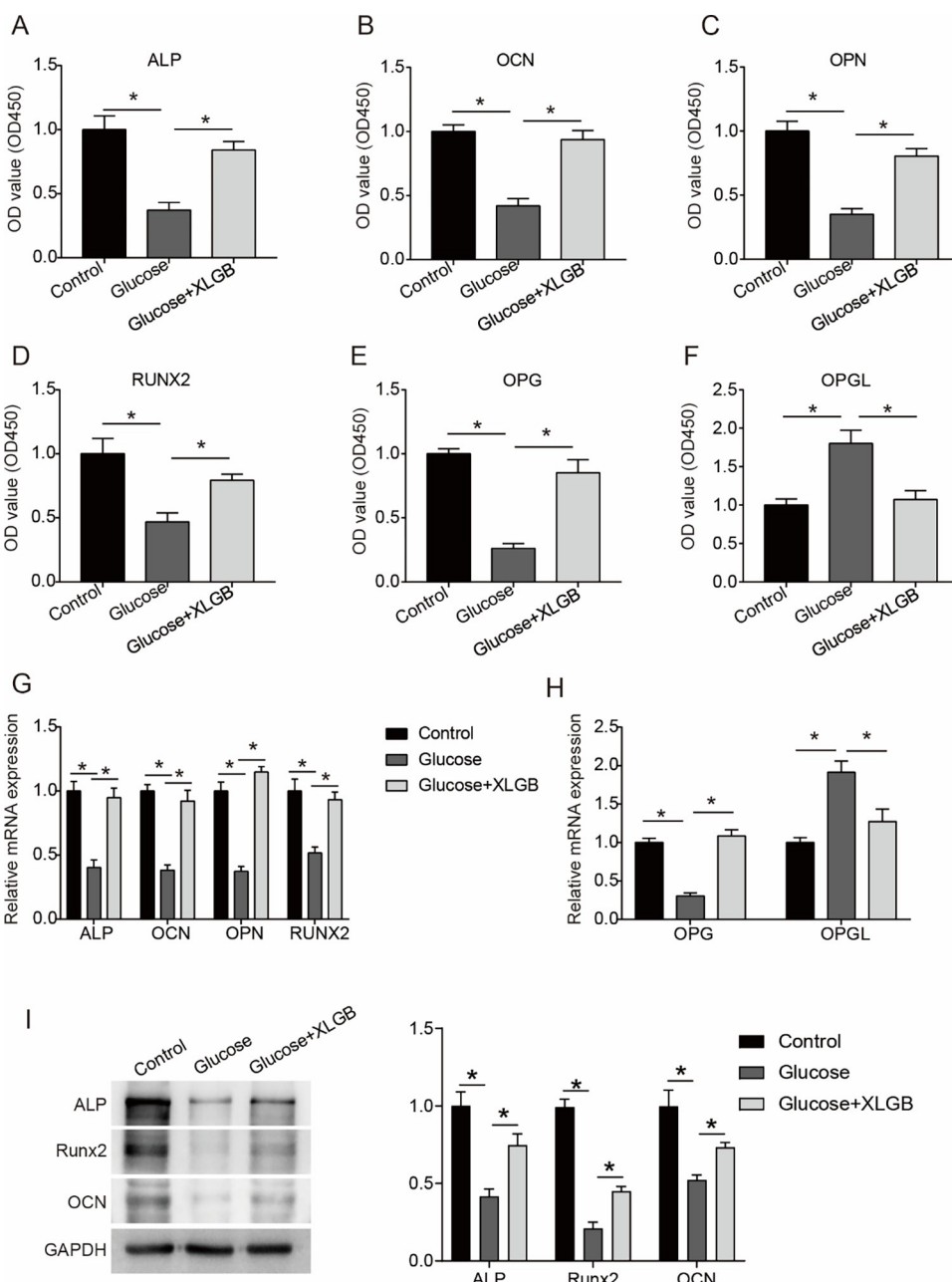

**Fig 4. XLGB rescues the reduced osteogenic differentiation-associated markers in the MG63 cells exposed to high glucose.** A-F. The protein levels of ALP (A), OCN (B), OPN (C), RUNX2 (D), OPG (E), and OPGL (F) were examined using ELISA assay in the MG63 cells incubated in control, high glucose and high glucose +XLGB mediums. G. The mRNA levels of ALP, OCN, OPN, and RUNX2 were examined by RT-PCR analysis in MG63 cells after indicated treatment. H. The mRNA levels of OPG and OPGL in MG63 cells. I. the levels of osteogenesis differentiation-associated markers were determined by western blot. Control, cells treated with mannitol; Glucose, cells treated with 20 mM of glucose; Glucose +XLGB, cells treated with 20 mM of glucose plus 200 mg/L of XLGB. Data represented mean ± SD of three separate experiments. *P<0.05.

associated markers induced by high glucose were significantly blocked by XLGB overexpression in MG63 cells (**Fig 4I**). Taken together, we confirmed that the inhibition effect of high glucose on the differentiation of MG63 cells could be rescued by XLGB.

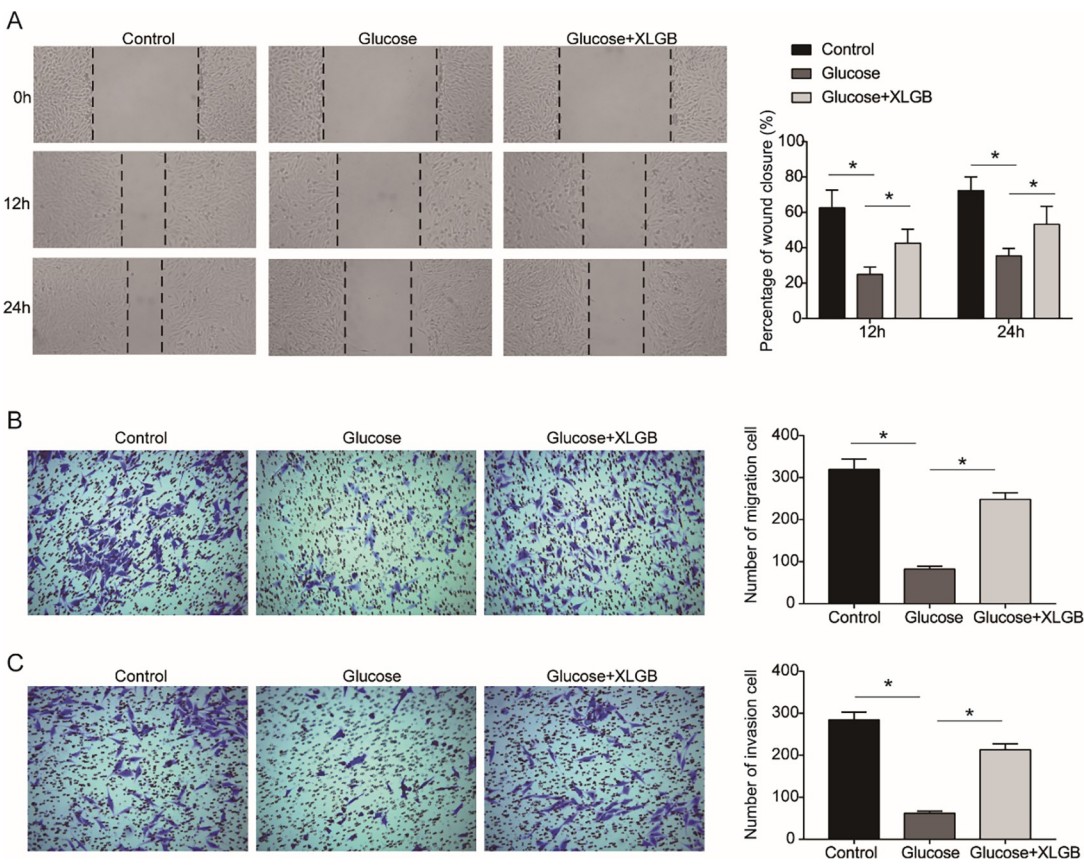

**Fig 5. XLGB rescues the reduced the migration and invasion abilities of MG63 cells caused by high glucose.** A. Cell migration was assessed by wound-healing assay in MG63 cells post treatments with control, high glucose and high glucose +XLGB. B, C. Transwell assay was performed to determine the migration (B) and invasion (C) abilities of MG63 cells. Control, cells treated with mannitol; Glucose, cells treated with 20 mM of glucose; Glucose +XLGB, cells treated with 20 mM of glucose plus 200 mg/L of XLGB. Data represented mean ± SD of three separate experiments. *P<0.05.

### XLGB rescues the MG63 cells against the effect of high glucose on migration and invasion

Wound-healing and transwell assays were performed to assess the effects of XLGB on the migration and invasion of MG63 cells induced by high glucose. As shown in **Fig 5A**, compared with the control group, the wound closure of MG63 cells was inhibited by high glucose at 12 or 24 h. We found that XLGB rescued the suppression of migration ability resulting from high glucose at 12 or 24 h (**Fig 5A**). The transwell results also confirmed that the reduction in migration ability of MG63 cells caused by high glucose could be restored by XLGB (**Fig 5B**). Moreover, high glucose also reduced the invasion ability of MG63 cells, which could be restored by XLGB (**Fig 5C**). Then, to verify that the effect of XLGB on MG63 cells was specific to high glucose treatment, we detected the effects of XLGB overexpression on the proliferation, migration and invasion of MG63 cells under conventional culture conditions. As shown in **Fig 6**, XLGB overexpression had no significant effect on the proliferation and migration of MG63 cells cultured in conventional conditions.

### Inhibition of the PI3K/Akt signaling pathway by high glucose is rescued by XLGB treatment

The PI3K/Akt signaling pathway has been reported to be inhibited by high glucose [20] and play an essential role in cellular functions, including cell proliferation, differentiation and

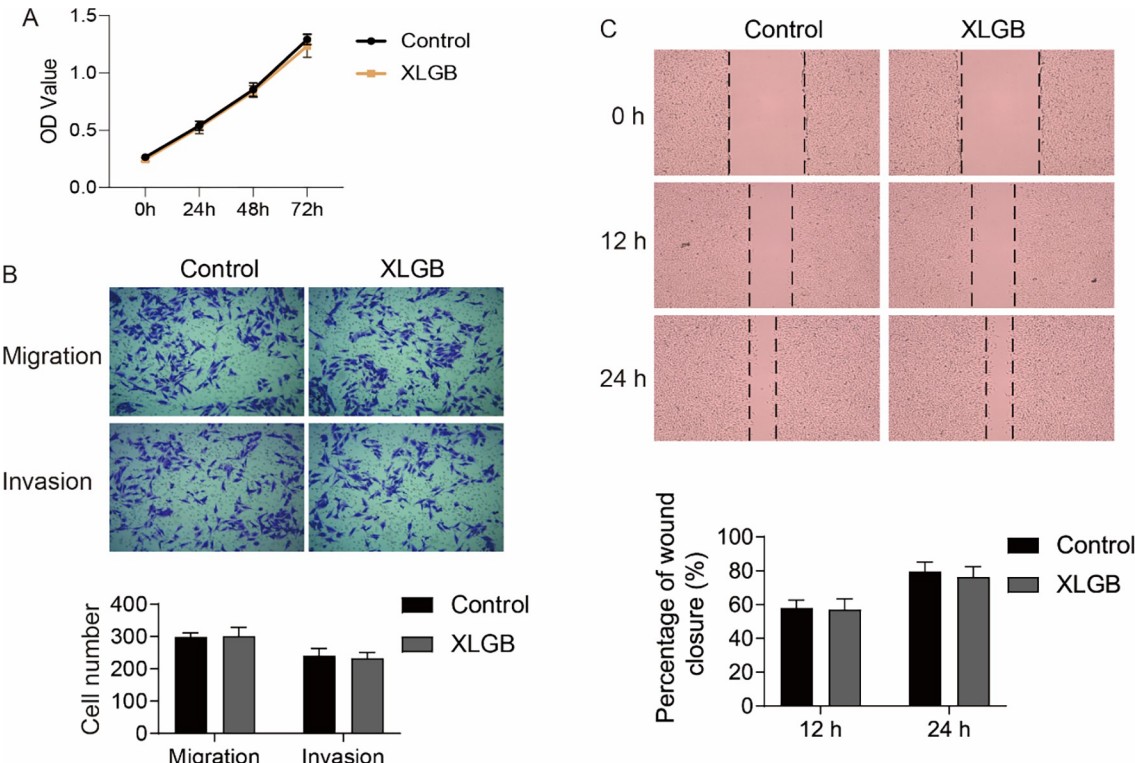

**Fig 6. Effects of XLGB overexpression on the proliferation, migration and invasion of MG63 cells under conventional culture conditions.** A. The proliferation was determined by CCK8 assay in control and XLGB overexpressed MG63 cells. B. Wound-healing assay was performed to detect cell migration. C. Transwell assay was performed to detect the migration and invasion of MG63 cells. Data represented mean ± SD of three separate experiments.

apoptosis. To investigate the possible involvement of the PI3K/Akt signaling pathway in the XLGB rescuing MG63 cells against the effect of high glucose, the expressions of key components of the PI3K/Akt signaling pathway were examined using western blot. It was indicated that high glucose significantly inhibited the phosphorylation level of Akt (p-Akt) and the expression of p70S6K in MG63 cells (**Fig 7A**). In comparison, XLGB could attenuate these downregulations in MG63 cells (**Fig 7A**). In addition, the downregulated expression levels of

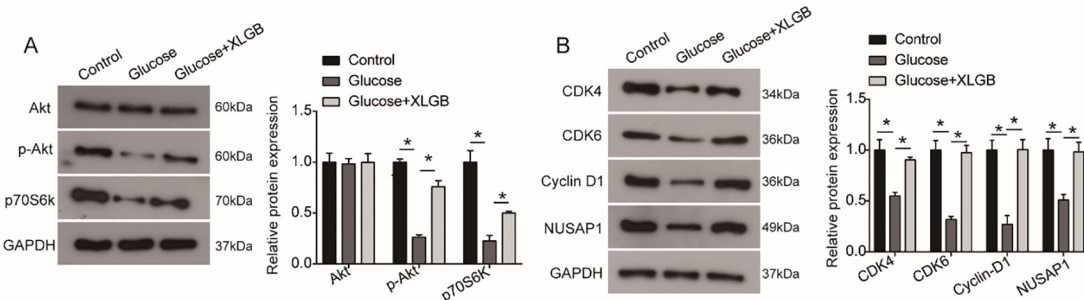

**Fig 7. XLGB rescues the PI3K/Akt signaling pathway in MG63 cells down-regulated by high glucose.** A. Western blot analysis was used to examine the protein levels of key components of the PI3K/Akt signaling pathway in MG63 cells after indicated treatment. B. The protein levels of cell cycle-related proteins (CDK4, CDK6, Cyclin D1) were detected by western blot analysis. Control, cells treated with mannitol; Glucose, cells treated with 20 mM of glucose; Glucose +XLGB, cells treated with 20 mM of glucose plus 200 mg/L of XLGB. Data represented mean ± SD of three separate experiments. *$P < 0.05$.

cell cycle-related proteins, CDK4, CDK6, Cyclin D1, and NUSAP1 caused by high glucose were also rescued by XLGB in MG63 cells (**Fig 7B**).

Finally, we examined the effect of XLGB on osteogenesis differentiation-associated markers under high glucose treatment in MG63 cells. As shown in **Fig 7C**, high glucose significantly inhibited the levels of ALP, Runx2 and OCN in MG63 cells. In comparison, these inhibitions of differentiation markers were significantly blocked by XLGB overexpression in high glucose treated MG63 cells.

## Discussion

It is generally known that DM could increase the risk of osteoporosis and fragility fractures in patients [21,22]. It has been revealed that high glucose could inhibit the biological function of osteoblasts and monocytic cells *in vitro* [5,19,23]. In the present study, our data also demonstrated that high glucose inhibited the proliferation, migration and invasion abilities of MG63 cells, and promoted apoptosis, as well as down-regulated the expression of osteoblast differentiation-associated markers. Moreover, we found that XLGB, a traditional Chinese herbal formulation for the treatment of osteoporosis, could rescue MG63 cells against the effects of high glucose on the proliferation, migration, invasion, differentiation, and apoptosis of MG63 cells. Moreover, XLGB overexpression had no significant effect on the proliferation and migration of MG63 cells. However, whether the inhibitory effect of high concentration XLGB on the proliferation of MG63 cells is related to the solution environment needs to be further investigated in the future.

In addition to cell proliferation, the apoptosis of osteoclasts and osteoblasts is a key determinant in maintaining the balance of bone metabolism [24]. Li JL *et al.* have shown that high glucose could induce the apoptosis of osteoblast cell line MC3T3-E1 [25]. Our study demonstrated that high glucose promoted apoptosis of MG63 cells and down-regulated the expression of Bcl-2, while the expressions of Bax and active Caspase 3 were up-regulated. Importantly, XLGB could rescue the increase in apoptosis caused by high glucose, as well as the expression changes of these apoptosis-related proteins. It is well known that Bcl-2, Bax and active Caspase 3 exert a vital role in triggering apoptosis. Bcl-2/Bax ratio is the key to determine cell fate. Bcl-2 functions as an anti-apoptotic protein by binding and silencing Bax, while up-regulation of Bax can resist Bcl-2 and trigger apoptosis [26,27]. Active Caspase 3 is the key executor of apoptosis. Therefore, our results suggest that XLGB alleviates high glucose-induced apoptosis of MG63 cells by regulating the Bcl-2/Bax axis and Caspase 3 activation.

It has been reported that high glucose inhibits the osteoblast differentiation by regulating the STAT3/SOCS3 signaling pathway [19]. ALP is one of the most important early markers of osteoblast differentiation [28]. RUNX2 has been widely recognized as the first transcription factor for determination of osteoblast cell lineage, inducing mesenchymal stem cells to differentiate into immature osteoblasts [29]. RUNX2 can regulate the expression of osteoblast differentiation markers such as ALP, OPN and OCN [30]. OPG is produced by osteoblast lineage and increases with cell differentiation, inhibiting osteoclast formation, differentiation and maturation. OPGL, also known as RANKL, is the neutralizing ligand for OPG and promotes osteoclasts formation and differentiation. The OPG/OPGL system has been proven to be a key factor in osteoclast formation, differentiation and regulation of bone resorption [31]. Our research showed that high glucose inhibited the expression of ALP, OPN, OCN, RUNX2, and OPG in MG63 cells, and up-regulated the expression of OPGL; while XLGB could rescue their expression in MG63 cells. These results imply that XLGB can promote osteoblast differentiation in a high glucose environment.

As one of the most critical signaling pathways regulating cell proliferation, apoptosis and differentiation, the PI3K/Akt signaling pathway has been shown to be down-regulated in osteoporosis rats and involve in osteoblast differentiation and growth [30]. The PI3K/Akt signaling pathway is the center of the signaling pathway network that regulates the function of osteoblasts and osteoclasts, playing a critical role in maintaining the dynamic balance of bone tissue [32]. Therefore, the PI3K/Akt signaling pathway may function as a therapeutic target for osteoporosis treatment. Akt phosphorylated by activated PI3K can activate or inhibit downstream target proteins such as mTOR, Bad, Caspase 9, and GSK-3, participating in the regulation of cell survival, proliferation, apoptosis and differentiation [33]. p70S6K, a key downstream target of mTOR, functions in promoting cell proliferation by regulating translation initiation and protein synthesis. mTOR can promote the combination of Cyclin D1 with CDK4 and CDK6 to form a Cyclin 1-CDK4-CDK6 complex which triggers the cell cycle from the G1 phase to the S phase [34]. Akt can prevent the degradation of Cyclin D1 by inhibiting the kinase activity of GSK3β, regulating the cell cycle. Additionally, NUSAP1, an important microtubule and chromatin binding protein, is a cell cycle-related protein that has been reported to play a key role in mitosis and involve in proliferation [35]. Herein, our study found that XLGB could eliminate the inhibition of high glucose on the PI3K/Akt signal pathway in MG63 cells, and reverse the expression of p70S6K, CDK4/6, Cyclin D1 and NUSAP1. In addition, inhibitions of differentiation-associated markers induced by high glucose were significantly blocked by XLGB overexpression in MG63 cells. These data indicate that the PI3K/Akt signaling pathway is involved in the protective effects of XLGB on osteoblast differentiation and proliferation in the background of high glucose.

## Conclusions

In summary, this study revealed that XLGB could protect osteoblast-like cells from high glucose induced cells damage. Our data demonstrated that XLGB could not only rescue the inhibition of high glucose on osteoblasts proliferation, migration and differentiation, but also reduce the promoting effect of high glucose on osteoblast apoptosis. Therefore, the potential therapeutic mechanisms of XLGB might be partially attributed to inhibiting the osteoblast apoptosis and promoting the bone formation of osteoblasts. These findings might provide novel insights into the mechanism of XLGB in the treatment of osteoporosis. It has been identified that the extract of XLGB, XLBG-B is the key component in preventing bone loss in mice induced by OVX [18], but its precise bioactive components that protect the osteoblasts growth and differentiation require further research.

### The reference codes for the antibodies used for Elisa (Abcam)

alkaline phosphatase (ALP) ab254503

Osteocalcin (OCN) ab270202

Osteopontin (OPN) ab269374

runt-related transcription factor 2 (RUNX2) ab275220

Osteoprotegerin (OPG) ab189580

Osteoprotegerin ligand (OPGL) ab213841

### Primer sequences

|  | F (5'-3') | R (5'-3') |
|---|---|---|
| ALP | CATACACACATGCTCAGCGC | TGGTCAATTCTGCCTCCACC |
| OCN | GTGCAGCCTTTGTGTCCAAG | TCAGCCAACTCGTCACAGTC |
| OPN | AATACCCAGATGCTGTGGCC | CTGGCTGTCCACATGGTCAT |
| RUNX2 | GCGCATTCCTCATCCCAGTA | GGCTCAGGTAGGAGGGGTAA |

## Supporting information

**S1 Raw images.**
(PDF)

## Author Contributions

**Conceptualization:** Xiaoqian Zhang.

**Data curation:** Xinlong Chen, Xiaoqian Zhang.

**Formal analysis:** Xinlong Chen, Zhongwen Zhang, Xiaoqian Zhang.

**Funding acquisition:** Xiaoqian Zhang.

**Investigation:** Xinlong Chen, Yan Li, Zhongwen Zhang, Liping Chen, Yaqian Liu, Xiaoqian Zhang.

**Methodology:** Xinlong Chen, Zhongwen Zhang, Liping Chen, Shuhong Huang, Xiaoqian Zhang.

**Project administration:** Xiaoqian Zhang.

**Resources:** Xiaoqian Zhang.

**Software:** Xinlong Chen, Xiaoqian Zhang.

**Supervision:** Xiaoqian Zhang.

**Validation:** Xiaoqian Zhang.

**Visualization:** Xiaoqian Zhang.

**Writing – original draft:** Xinlong Chen, Yan Li, Xiaoqian Zhang.

**Writing – review & editing:** Xinlong Chen, Yan Li, Xiaoqian Zhang.

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
