## [Decision Letter · Decision Letter 0]

19 May 2022

PONE-D-21-39926Xianling Gubao attenuates high glucose-induced bone metabolism disorder in MG63 osteoblast-like cellsPLOS ONE

Dear Dr. zhang,

Thank you for submitting your manuscript to PLOS ONE. After careful consideration, we feel that it has merit but does not fully meet PLOS ONE’s publication criteria as it currently stands. Therefore, we invite you to submit a revised version of the manuscript that addresses the points raised during the review process.

In fact, your manuscript has been carefully evaluated by two independent subject experts, but both reviewers have raised various concerns on your manuscript, they have also mentioned to perform few western blotting experiments to substantiate your views.

We look forward to receiving your revised manuscript.

Kind regards,

Chandi C. Mandal, Ph.D.

Academic Editor

PLOS ONE

Journal Requirements:

Reviewers' comments:

Reviewer's Responses to Questions

5. Review Comments to the Author

Reviewer #2: 1. I request authors to do westren blot analysis for some of the osteogenesis differentiation-associated markers.

2. The effect of XLGB on MG63 alone should be studied to conclude that XLGB specifically rescue the MG63 from High Glucose treatment from osteoblasts proliferation, migration. differentiation etc and inhibiting the osteoblast apoptosis .

---

## [Author Response · Author response to Decision Letter 0]

28 Jul 2022

Dear Editor,

Thank you for reading and reviewing our manuscript. Those comments are all valuable and very helpful for revising and improving our paper. According to the editor and reviewers’ comments, we have made extensive modifications to our manuscript to make our results convincing, and we wish it to be considered for publication in your journal. The detailed point-by-point responses are listed below. 

Reviewer #2: 1. I request authors to do western blot analysis for some of the osteogenesis differentiation-associated markers.

Response: Thank you very much for the comments. Thank you very much. We detected differentiation-associated markers ALP, Runx2 and OCN. The results were showed in Figure 7C.

2. The effect of XLGB on MG63 alone should be studied to conclude that XLGB speciffcally rescue the MG63 from High Glucose treatment from osteoblasts proliferation, migration, differentiation etc and inhibiting the osteoblast apoptosis.

Response: We detected the effect of XLGB on MG63 alone, and the results were showed in Figure 6.

Thank you very much for your reconsidering our revised manuscript for potential publication. According to the reviewer’s comments, we have improved the manuscript extensively. If there are any other modifications we could make, we would like very much to modify them and we really appreciate your help. We look forward to hearing from you regarding our submission. We would be glad to respond to any further questions and comments that you may have. 

Sincerely yours

Xiaoqian Zhang

---

## [Decision Letter · Decision Letter 1]

5 Oct 2022

Xianling Gubao attenuates high glucose-induced bone metabolism disorder in MG63 osteoblast-like cells

PONE-D-21-39926R1

Dear Dr. zhang,

We’re pleased to inform you that your manuscript has been judged scientifically suitable for publication and will be formally accepted for publication once it meets all outstanding technical requirements.

Kind regards,

Gary S. Stein

Academic Editor

PLOS ONE

Additional Editor Comments (optional):

Reviewers' comments:

Reviewer's Responses to Questions

**Comments to the Author**

1. If the authors have adequately addressed your comments raised in a previous round of review and you feel that this manuscript is now acceptable for publication, you may indicate that here to bypass the “Comments to the Author” section, enter your conflict of interest statement in the “Confidential to Editor” section, and submit your "Accept" recommendation.

Reviewer #1: All comments have been addressed

Reviewer #2: (No Response)

2. Is the manuscript technically sound, and do the data support the conclusions?

Reviewer #1: Yes

Reviewer #2: Partly

3. Has the statistical analysis been performed appropriately and rigorously? 

Reviewer #1: Yes

Reviewer #2: I Don't Know

4. Have the authors made all data underlying the findings in their manuscript fully available?

Reviewer #1: Yes

Reviewer #2: No

5. Is the manuscript presented in an intelligible fashion and written in standard English?

Reviewer #1: Yes

Reviewer #2: Yes

6. Review Comments to the Author

Reviewer #1: Hi,

No further comment from all sides.

All issues were addressed very well, especially the immunoblotting of osteogenic markers.

I

Thank you

Reviewer #2: (No Response)

7. PLOS authors have the option to publish the peer review history of their article (what does this mean?). If published, this will include your full peer review and any attached files.

Reviewer #1: No

Reviewer #2: No

---

## [Editor Report · Acceptance letter]

9 Dec 2022

PONE-D-21-39926R1 

Xianling Gubao attenuates high glucose-induced bone metabolism disorder in MG63 osteoblast-like cells 

Dear Dr. Zhang:

I'm pleased to inform you that your manuscript has been deemed suitable for publication in PLOS ONE. Congratulations! Your manuscript is now with our production department. 

Kind regards, 

on behalf of

Dr. Gary S. Stein 

Academic Editor

PLOS ONE